# Update of Natural Products and Their Derivatives Targeting Epstein–Barr Infection

**DOI:** 10.3390/v16010124

**Published:** 2024-01-15

**Authors:** Rosamaria Pennisi, Paola Trischitta, Marianna Costa, Assunta Venuti, Maria Pia Tamburello, Maria Teresa Sciortino

**Affiliations:** 1Department of Chemical, Biological, Pharmaceutical and Environmental Science, University of Messina, Viale Ferdinando Stagno d’Alcontres 31, 98166 Messina, Italy; paola.trischitta@studenti.unime.it (P.T.); marianna.costa@studenti.unipg.it (M.C.); maria.tamburello1@studenti.unime.it (M.P.T.); 2Department of Chemistry, Biology and Biotechnology, University of Perugia, Via Elce di Sotto 8, 06123 Perugia, Italy; 3International Agency for Research on Cancer (IARC), World Health Organization, 69366 Lyon, CEDEX 07, France; assuntavenuti@gmail.com

**Keywords:** Epstein–Barr virus, natural antivirals, human oncogenic virus

## Abstract

Epstein–Barr (EBV) is a human γ-herpesvirus that undergoes both a productive (lytic) cycle and a non-productive (latent) phase. The virus establishes enduring latent infection in B lymphocytes and productive infection in the oral mucosal epithelium. Like other herpesviruses, EBV expresses its genes in a coordinated pattern during acute infection. Unlike others, it replicates its DNA during latency to maintain the viral genome in an expanding pool of B lymphocytes, which are stimulated to divide upon infection. The reactivation from the latent state is associated with a productive gene expression pattern mediated by virus-encoded transcriptional activators BZLF-1 and BRLF-1. EBV is a highly transforming virus that contributes to the development of human lymphomas. Though viral vectors and mRNA platforms have been used to develop an EBV prophylactic vaccine, currently, there are no vaccines or antiviral drugs for the prophylaxis or treatment of EBV infection and EBV-associated cancers. Natural products and bioactive compounds are widely studied for their antiviral potential and capability to modulate intracellular signaling pathways. This review was intended to collect information on plant-derived products showing their antiviral activity against EBV and evaluate their feasibility as an alternative or adjuvant therapy against EBV infections and correlated oncogenesis in humans.

## 1. Introduction

EBV commonly infects people in developed and developing countries. Most cases are asymptomatic, although infectious mononucleosis can manifest in individuals, particularly as the age of infection increases. EBV infects epithelial cells first, then spreads to B cells in lymphoid tissue, establishing lifelong latent infection [1]. Immortalized B cells with EBV have oncogenic potential, leading to cell cycle progression and transformation events linked to several cancers [2]. While T cells usually regulate EBV proliferation, immunocompromised individuals may lack this control, associating latent EBV infection with cancers. No specific FDA or EMA-approved anti-EBV drug exists, but a combination of standard antivirals and immunomodulators has shown effectiveness. However, accumulating epidemiological, serological, and virological evidence substantiates the involvement of EBV in the etiology of multiple sclerosis (MS). Recent extensive population-based studies provide compelling evidence that EBV infection is probably a prerequisite for developing the disease [3]. Therefore, discovering novel, potent, and safe antiviral agents targeting virus particles and cell response to viral infection remains a daunting challenge. The therapeutic use of natural compounds to treat various diseases dates back to ancient times, and more formulations are still used today as supportive medicine [4]. Their activity is attributable to secondary metabolites isolated and characterized as bioactive compounds. Their antiviral activity was an important revelation because viral infections and the long-term prevalence of drug resistance remain a worldwide problem that cannot be underestimated. Numerous natural compounds have been screened and identified as inhibitors targeting various steps of viral replication, such as the entry, uncoating, genome replication, late gene expression, assembly, exit, and cellular processes required for virion production [5]. The advantage of their use lies in the low toxicity and availability of the products and in the lack of drug resistance phenomena. At the same time, many natural compounds and pure metabolites exhibited potent inhibitory activity in vitro, but they were much less effective in vivo due to absorption and metabolic modification phenomena [6]. However, natural products remain the best resource for future development as potent and safe antiviral formulations. The review summarizes EBV pathogenesis, reactivation, and its role in related diseases, providing updates on natural inhibitors targeting the EBV lytic cascade and cellular pathways associated with oncogenesis.

## 2. EBV-Associated Disorders and Antiviral Therapy

EBV, discovered over 50 years ago, causes latent infection, efficiently infecting the nasopharyngeal mucosa and B lymphocytes and leading to their immortalization. In immunocompetent individuals, most infected cells are eliminated by cytotoxic lymphocytes after acute infection, with a small percentage of B cells transitioning through different latent states (pre-latent, latency III, and latency I). Reactivation of the infection involves an interplay between lytic and latent cycles, contributing to EBV-associated diseases [7]. EBV-associated carcinogenesis is a complex process, activating oncogenes encoded by the virus. It transforms B-lymphocytes into lymphoblastoid cell lines (LCLs), expressing latent genes and microRNAs, evolving in carcinogenesis. Proteins like LMP1 and LMP2A activate signaling pathways in cell cycle progression. EBNA proteins, including EBNA2, EBNA3A, and EBNA3C, manipulate cell cycle regulators, inhibit apoptosis, and modulate immune responses, contributing to oncogenesis. The BHRF1 miRNA is implicated in B cell transformation, while EBNA-LP cooperates with EBNA2 in expressing viral oncogenes. EBV is associated with several cancers, each exhibiting different latency programs. Burkitt lymphomas (BL), nasopharyngeal carcinoma (NPC), gastric carcinoma (GC), and Hodgkin’s lymphoma (HL) express specific latency programs [8]. The virus is linked to MS, with cross-reactive antibodies suggesting molecular mimicry as a mechanism for MS pathology [3].

Therapeutic strategies for EBV-associated malignancies include antivirals, lytic cycle induction, small molecule inhibitors, immunostimulators, and vaccines. Immune checkpoint therapies show promise. Effective antivirals remain elusive, but small molecule inhibitors targeting EBNA1, such as VK-2019, are promising and undergoing clinical trials for advanced nasopharyngeal carcinoma. Adoptive cell therapy involving engineered T-cell receptors and CAR T cell therapy for EBV proteins is under investigation [9,10,11,12,13].

The absence of FDA-approved vaccines underscores the need for prophylactic measures. Various vaccines are explored, including recombinant envelope protein, live recombinant, and mRNA vaccines [14,15]. Drugs like JQ1, cimetidine, and antiretrovirals show anti-EBV activity by presenting different approaches to inhibit EBV replication [16,17,18,19,20,21,22]. Further research and clinical studies are crucial for validating efficacy and safety across diverse patient populations.

### 2.1. EBV Entry in B and Epithelial Cells

The EBV virion, with a diameter of about 150–170 nm, is composed of an icosahedral nucleocapsid with 162 capsomers surrounded by an envelope. The viral genome comprises about 170 kb of double-stranded DNA [8]. Viral surface glycoproteins are responsible for recognizing and binding to cellular receptors and consequent membrane fusion. EBV possesses a wide range of glycoproteins, such as gp350, gp42, gH, gL, and gB [23]. Since the entry mechanism of EBV is much more sophisticated, many receptors involved in infecting different cell types have been identified. During primary infection, the virus crosses the mucosal epithelial cells by transcytosis and then infects B cells in the submucosal secondary lymphoid tissues [24]. The different cellular targets allow differences between the mechanisms underlying EBV attack on B lymphocytes and epithelial cells. Indeed, B lymphocytes express CD21/CD35 receptors, also known as complement receptor type 2 and complement receptor type 1 (CR2/CR1), mainly expressed in lymphocytes [25,26,27]. The interaction between the EBV envelope glycoprotein, gp350/220, and CD21/CD35 is responsible for the binding and triggers a signaling cascade, allowing the penetration of the virion in B cells [26]. Thus, gH, gL, and gp42 form a heterotrimer that binds to human leukocyte antigen (HLA) class II molecules (HLA-II) on B cells, leading to a conformational change and to the activation of gB, which plays a direct role in membrane fusion [23,28,29,30,31,32,33]. Unlike B cells, epithelial cells do not constitutively express CD21 or CD35. Thus, the binding is guaranteed by the interaction of EBV glycoprotein BMRF2 with integrins β1, α5, and α3 [34], and the consequent entry of virion involves various cellular receptors which interact with gH/gL or gB [35]. BBRF2 is an EBV tegument protein with putative homologs in all three herpesvirus subfamilies [36]. The interaction between BBRF2 and BSRF1 tegument proteins prevents BSRF1 degradation and increases viral infectivity [35]. Neuropilin-1 (NRP1) is a multifunctional protein that acts as a cellular entry cofactor for different viruses [37]. NRP1 has been reported to interact with gB and mediate the EBV infection. Ephrin receptor 2 type A (EphA2), which belongs to the largest family of receptor tyrosine kinases, has been recognized as a receptor for many pathogens and has been shown to bind gH/gL complex during the infection of epithelial cells by EBV [38,39,40]. Following viral attachment to cellular receptors, the virion internalizes and delivers its epigenetically naïve linear DNA genome to the nucleus [41]. The circularization of the viral DNA is one of the first events useful to protect the DNA ends from degradation and minimize induction of the DNA damage response of the infected cell. Many lytic and latent viral genes are expressed during the first hours of infection. Approximately ten days later, EBV latent gene expression predominates, and lytic gene expression becomes almost undetectable. In this scenario, the latency phase is established. Under the influence of appropriate stimuli, the expression of virus-encoded transcriptional factors, BZLF-1 and BRLF-1, activates a productive gene expression pattern in the replication of the viral genome and progeny virus genesis (Figure 1) [42].

### 2.2. The Role of EBV Proteins in Cell Cycle Progression and Oncogenesis

EBV is associated with specific human cancers such as Burkitt’s lymphoma, nasopharyngeal carcinoma, and gastric cancer [8]. These tumors express latent EBV antigens and a viral genome synchronized with host genome replication, dependent on chromosomal initiation factors ORC2 and Cdt1 [42,43,44]. While EBV in cancer cells is mainly in the latent state, the lytic cycle of the virus contributes to tumor development by promoting inflammation and angiogenesis via the secretion of cytokines and growth factors like IL-10, IL-8, TGF-β, and VEGF [45,46]. During lytic cycle reactivation, the immediate-early (IE) lytic genes, BZLF1 and BRLF1, are expressed, activating early genes and EBV genome replication with a rolling-circle mechanism [45,47]. EBV-associated carcinogenesis is a multistep process in which oncogenes encoded by EBV play a crucial role. The virus transforms human B-lymphocytes into LCLs expressing various latent genes, including Epstein–Barr virus nuclear antigens (EBNA 1, 2, 3A, 3B, 3C, and EBNA leader protein), latent membrane proteins (LMP-1 and LMP-2), small RNAs (EBER1 and EBER2), and microRNAs [8]. LMP1 and LMP2A activate signaling pathways involved in cell cycle progression, including NF-κB, JNK, p38 MAPK, JAK/STAT, and PI3K/Akt [48,49,50,51]. The carboxy-terminal activating region 1 (CTAR1) of LMP1 induces the expression of EGFR and TRAF1, promoting B cell proliferation and differentiation by deregulating CDK2 and Rb, involved in G1/S cell cycle progression [52]. EBNA 3A/C inhibits the transcription of CDK inhibitors, p14ARF and p16INK4A, neutralizing the tumor suppressor gene Rb and maintaining constitutive cell cycle activation [53,54]. It also directly interacts with the C-terminus region of p53 by modulating its transcriptional and apoptotic activities [55]. The inactivation of the BHRF1 miRNA results in B cell transformation and LCL growth, suggesting its role in oncogenesis [56,57,58,59,60,61]. EBNA2 induces transcription of the cellular oncogene MYC and impairs EBV lytic replication by inducing expression of the methylcytosine dioxygenase Tet 2 (TET2), blocking methylation sites for BZLF1 binding [62,63,64]. The EBNA leader peptide (EBNA-LP) cooperates with EBNA2 in expressing viral oncogenes, including LMP1 [65]. EBNA3A and EBNA3C rescue infected cells driven into a proliferative state by EBNA2-dependent MYC expression by negatively regulating pro-apoptotic proteins BIM and p16INK4a [66], preventing the switch to lytic replication by repressing BLIMP1 expression [67]. EBNA3B ensures sufficient immune cell infiltration among EBV-transformed B cells to limit lymphoma development [68]. EBV-associated carcinogenesis involves immunosuppressive conditions, HIV-1 co-infection and transplantation [69], the activation of the inflammatory system, and genetic or epigenetic predisposition and alterations in the host genome [70]. The virus induces B cells to become activated lymphoblasts by differentiating into resting memory B cells where the virus persists. This differentiation occurs via the germinal center (GC) reaction, representing a high-risk region for genetic instability and antiapoptotic signals leading to B-cell lymphoma [71,72].

## 3. Natural Therapeutic Compounds Targeting EBV Infection

The categorization of antiviral agents into virucides, chemotherapeutic agents, and immunomodulators is a common classification based on their mechanisms of action. Virucides are substances or agents that directly inactivate or destroy viruses. They act by disrupting the viral structure or interfering with essential viral functions. Otherwise, antiviral chemotherapeutic agents are drugs designed to target viral replication processes specifically. These drugs interfere with the virus’s ability to replicate or spread within the host. Several antiviral drugs may target various stages of the viral life cycle, such as viral entry, genome replication, or virion release. Lastly, immunomodulators are substances that modulate or regulate the immune system’s response to viral infections. They can enhance or suppress immune functions to achieve a balanced and effective antiviral response. Immunomodulators are often used to treat viral infections by either boosting the immune system’s ability to fight the virus or preventing excessive immune responses that can lead to inflammation and tissue damage [73]. It is important to note that these categories are not mutually exclusive, and some antiviral agents may exhibit properties of more than one category. Several natural compounds have been studied for their potential antiviral properties. It is important to note that while these compounds may exhibit antiviral activity in in vitro studies, their effectiveness in treating viral infections in humans may differ. Here are some examples of natural compounds with reported antiviral properties (Figure 2).

### 3.1. Natural Products Targeting EBV Binding

Viral entry generally occurs either through direct fusion of the virus with the surface membrane or by endocytic uptake [74]. Although the viral attachment to host cells represents a significant phase for productive infection, low numbers of compounds isolated from medicinal plants are known to inhibit the early stages of EBV infection (Table 1). It was reported that the quercetin isolated from licorice interferes with the recognition of EBV receptors such as CD21, CD35, and HLAII in AGS cells) and consequently blocks EBV entry [75]. In addition, the treatments of quercetin or isoliquiritigenin limit EBV infection in coinfected gastric adenocarcinoma cells and lymphoblastoid cells containing EBV. Further, it was reported that glycyrrhizic acid (GL), a component of licorice root (*Glycyrrhizae radix*), is active against EBV replication in superinfected Raji cells in a dose-dependent manner and interferes with the early step of EBV replication cycle [76]. The potential role of genipin as a natural crosslinker for proteins and its impact on the interaction between EBV attachment proteins and cellular receptors is an interesting hypothesis [77]. Genipin may bind to EBV attachment proteins such as gp350 and gp42. These viral proteins are involved in the initial stages of EBV infection, particularly in attaching to host cells. If genipin interferes with the proper function of these proteins, it could disrupt the ability of the virus to attach to host cells. Further, extracellular genipin may bind to cellular EBV receptors such as CD21 and CD35. Doing so might block these cellular receptors from interacting with EBV attachment proteins. This interference prevents the virus from effectively attaching and entering into host cells [78]. In silico analysis underscores the potential of bruceantin, belonging to the family of triterpenes, and epigallocatechin-3-gallate (EGCG), major green tea catechin, as antiviral agents targeting the gH protein of EBV [79]. EGCG is known to influence the properties of viral envelopes in other viruses, such as influenza and herpes simplex virus (HSV) [80] and reduce the attachment of CHIKV and HCV to target cells [81,82]. These data suggest a broad-spectrum antiviral potential for EGCG, impacting various stages of the viral life cycle. Borenstein et al., 2020 demonstrated that ginkgolic acid, at a concentration of 100 µM, effectively prevented EBV membrane fusion. In particular, it has been described as an inhibition of the viral fusion glycoprotein gB that prevents EBV reactivation [83].

### 3.2. Natural Extracts Targeting EBV Lytic Infection and Oncogenesis

During the lytic cycle, EBV produces infectious virions, which can infect new cells. This mechanism can contribute to the spread of the virus within the host and potentially facilitate the infection of new target cells [84]. The ability of the virus to generate infectious particles increases the likelihood of establishing persistent infections. Further, the lytic phase involves the expression of various viral proteins, interfering with host cell signaling pathways, cell cycle, and cell survival. Some of these proteins may act as effectors contributing to oncogenesis. The role of EBV infection in tumorigenesis is a complex and multifaceted phenomenon. While it is true that lytic infection can lead to cell death, evidence suggests that the lytic phase of EBV may also promote oncogenesis via different mechanisms [85]. Specific viral proteins may also play a role in immune evasion, allowing infected cells to escape detection and elimination by the host immune system. Finally, lytic infection can induce an inflammatory response, and therefore, chronic inflammation is a well-known factor in promoting tumorigenesis [86]. Additionally, the lytic phase may influence the cellular microenvironment, creating conditions favorable for cell transformation and tumor growth. Consequently, a growing interest is being given to identifying compounds, specifically from natural extracts, that can effectively inhibit EBV lytic replication and block tumor development (Table 2).

The increased production of free radicals during EBV infection can lead to a radical chain reaction known as lipid peroxidation, causing damage to cell membranes and lipoproteins [87]. This cytotoxic and mutagenic phenomenon is associated with oxidative stress triggered by EBV infection [88,89]. Oxidative stress is related to the secondary lipid peroxidation products, such as malondialdehyde (MDA) and conjugated dienes (DC), generated by the decomposition of long-chain polyunsaturated fatty acids [90]. These products activate the transcription factors, including STAT3 and NF-κB [91]. Certain natural compounds have been found to reduce intracellular oxidative stress induced by in vitro treatment with TPA (12-O-tetradecanoylphorbol-13-acetate), leading to the inhibition of EBV replication. A study demonstrated that the treatment with TPA (8 nM) and extracts from *Olea europaea* L. var. *sativa* caused a significant decrease in MDA and DC levels in Raji cells, showing a protective effect against the induction of the EBV lytic cycle [87]. *Eugenia uniflora* extracts were assessed for their inhibitory effect on purified EBV DNA polymerase induced by phorbol 12-myristate 13-acetate (PMA) [92]. The study identified four principal compounds: gallocatechin, oenothein B, eugeniflorin D1, and eugeniflorin D2. The compounds showed varying degrees of inhibition, with eugeniflorins D1 and D2 exhibiting higher activity against EBV DNA polymerase than gallocatechin and oenothein B. The IC_50_ values of eugeniflorins D1 and D2 were lower than phosphonoacetic acid ones (PAA), suggesting their effectiveness in inhibiting EBV DNA synthesis. Nomura et al., 2002 synthesized polyphenol esters composed of gallic acid and ferulic acid, which demonstrated potent suppression of TPA-induced EBV activation at a concentration of 20 µM in vitro [93]. Zhang et al. [94] assessed the inhibitory effect of chlorogenic acid, protocatechuic acid, and gallic acid isolated from *Ficus hispida* L.f. fruits against TPA-induced EBV early antigen (EBV-EA) activation in Raji cells. These compounds exhibited inhibitory actions against EBV-EA activation, with IC_50_ values of 340, 481, and 473 mol ratio/32 pmol TPA [95]. The flavonoid-type compounds luteolin-7-O-β-D-glucopyranoside and apigenin-7-O-[β-D-apiofuranosyl (1→6)-β-D-glucopyranoside], isolated from *Lindernia crustacea* (L.) F.Muell. (Scrophulariaceae) effectively inhibited EBV lytic cycle [96]. In particular, the luteolin-7-O-β-D-glucopyranoside inhibits EBV lytic cycle at 20 μg/mL concentration. The inhibitory effect was associated with the downregulation of replication and the transcription activator (Rta) expression. Unlike, apigenin-7-O-[β-D-apiofuranosyl (1→6)-β-D-glucopyranoside] completely suppressed EBV virion production at a concentration of 50 µM. It inhibited EBV reactivation in the lytic cycle by suppressing the activities of the immediate-early gene Zta (BZLF 1) and Rta promoters [97]. Zta and Rta are essential proteins initiating the EBV lytic cycle [98,99]. Epigallocatechin-3-gallate (EGCG) has been observed to effectively block EBV lytic replication within a concentration range of 0.5 to 50 µM [100]. The inhibition of EBV lytic replication by EGCG has been demonstrated by (i) downregulation of LMP1 expression [100]. (ii) Inhibition of MAPKs/wt-p53 Signal Axis (in AGS-EBV cells) [100]; (iii) Inhibition of JNKs/c-Jun Signal Axis (in p53 mutant B95.8 cells) [100]. These findings suggest that EGCG exerts its inhibitory effect on EBV lytic replication by targeting LMP1 and modulating specific signaling pathways (MAPKs/wt-p53 and JNKs/c-Jun). The ability of EGCG to interfere with these molecular pathways highlights its potential usage as a therapeutic agent against EBV-associated diseases [98,99]). The study of protoapigenone and its analog protoapigenone 1′-O-isopropyl ether indicated their potential as selective and effective inhibitors of the EBV-lytic replication in EBV-positive Burkitt’s lymphoma (P3HR1) cells by impeding the expression of Rta protein [101]. Protoapigenone 1′-O-isopropyl ether was more selective against EBV and less toxic to the cells, making it a promising candidate for further investigation and development as a potential therapeutic agent for EBV-associated diseases, particularly in the context of Burkitt’s lymphoma [101]. The selectivity of a compound is crucial in developing effective and safe treatments for viral infections. Further research and clinical studies are necessary to explore these compounds’ full potential and safety profile in treating EBV infections [98,99,101,102]. Neo-clerodane diterpenoids from *Scutellaria barbata* [103] and from *Euphorbia milii* were reported to have potential antiviral activity. In particular, the acetone extract of *E. milii* inhibited the EBV lytic cycle. Thirteen new entrosane-type diterpenoids (1–13) were isolated from the *E. milii* and were evaluated against EBV. Among those, one derivative showed the most potent inhibitory activity with an EC_50_ value of 5.4 μM compared to the positive control (+)-rutamarin (EC_50_ = 5.4 μM) [99,104]. Lignans, isolated from *Saururus chinensis* and *Litsea verticillate* exhibited an antiviral effect against EBV via inhibition of the lytic cycle along with other biological activity. Among 28 lignans isolated from *S. chinensis*, manassantin B [99,105,106] demonstrated efficacy in blocking the lytic replication of EBV with lower cytotoxicity [107]. It has been shown that it targets BZLF1 gene expression by interrupting the AP-1 signal transduction. Further, it blocks the rapamycin complex 2 (mTORC2)-mediated phosphorylation of AKT Ser/Thr protein kinase at Ser-473, inhibits protein kinase Cα (PKCα) phosphorylation at Ser-657 and interrupts the mTORC2-PKC/AKT signaling pathway. Manassantin B’s ability to interfere with the mTORC2 pathway and AP-1 signal transduction suggests its potential as an antiviral agent against EBV. By targeting specific molecular pathways involved in the lytic replication of the virus, manassantin B may help suppress the expression of key genes like BZLF1, ultimately inhibiting the production of infectious virions [99,107].

The sulfated polysaccharides found in microalgae have also been reported to have antiviral activity. For instance, the methanol extracts of *Synechococcus elongatus* and *Ankistrodesmus convolutus* were reported to have low cytotoxicity and a strong antiviral effect against EBV in Burkitt’s lymphoma cells. The antiviral activity was measured by reducing the cell-free EBV DNA [99,108]. Moronic acid found in *Rhus chinensis* and *Brazilian propolis* inhibited the expression of Rta, Zta, and an EBV early protein. It reduces the ability of Rta to activate a promoter containing a Rta-response element. Since the expression of many EBV lytic genes depends on Rta, the treatment of P3HR1 Burkitt’s lymphoma cells with moronic acid substantially reduces the production of EBV particles by inhibiting the lytic cycle [99,109]. *Astragalus membranaceus* extract (thanks to its polysaccharides) inhibits EBV lytic cycle by suppressing the expression of the immediate–early protein, including Zta, Rta, and EA-D [99,110]. Henna (*Lawsonia inermis* L.) leaf powder and its primary pigment, lawsone (2-hydroxy-1,4-naphthoquinone), showed significant inhibition (>88%) of EBV-early antigen activation in vitro [99,111]. De Leo et al. demonstrated that resveratrol, a natural phenolic compound found in many plants and fruits, strongly induced apoptosis of EBV-positive Burkitt’s lymphoma cells, depending on the viral latency program. Additionally, resveratrol inhibited EBV reactivation by suppressing the lytic gene expression, including Rta, Zta, and EA-D. The production of virion was also reduced in a dose-dependent manner under resveratrol treatment [99,112,113]. Further, the studies conducted by Lee et al. in 2015 and Hwan Hee Lee et al. in 2016 provide insights into the distinct effects of quercetin and isoliquiritigenin on EBV infection and associated cancer. Quercetin showed higher antiviral activity than isoliquiritigenin. Indeed, it plays an important role in producing EBV progeny viruses from SNU719 cells, upregulates EBV lytic genes such as BZLF1, BRLF1, and BLLF1, and enhances the frequency of Fp (F promoter) usage in EBV. Moreover, the abrupt release of a large quantity of EBV progenies triggers apoptosis. Isoliquiritigenin, on the other hand, significantly upregulates EBV latent genes such as LMP1, LMP2, EBNA3A, and EBNA1, clearly contributing to the maintenance of EBV latency in SNU719 cells [75,114]. The ethanolic extract of *Andrographis paniculata*, and the compound of interest, andrographolide (at a non-toxic concentration for P3HR1 cells of 5 µg/mL), showed an antiviral effect against EBV via a mechanism of inhibition that occurs via blocking the transcription of the immediate–early genes that encode lytic proteins Rta and Zta [99,115]. 

The ethanolic extract of *Polygonum cuspidatum* inhibits the transcription of EBV’s immediate early genes and the expression of lytic proteins Rta, Zta, and EA-D. The primary active components in *P. cuspidatum* were reported to be resveratrol and emodin. The effective concentration of emodin required to inhibit the expression of immediate–early protein by 50% (EC50) obtained from flow cytometry was 4.83 μg/mL (17.87 μM), and its EC50 value to reduce DNA replication was 1.2 μg/mL [99,116,117,118]. Cordycepin, an adenosine derivative found in cordyceps (genus of Ascomycete fungi), has a similar chemical structure to adenosine; it can be intercalated into RNA molecules and can terminate RNA synthesis. It was reported to downregulate most EBV genes significantly by reducing EBV genome copy number by up to 55% in response to 125 µM cordycepin treatment, significantly lowering LMP2A and EBNA1 in SNU719 cells. Furthermore, cordycepin significantly suppressed EBV transmission from cell to cell in a coculture [99,119].
viruses-16-00124-t002_Table 2Table 2Natural compounds targeting EBV lytic infection.PlantSubstanceClassEBV TargetReferences*Olea europaea* L. var. *sativa*-
MDA and DCBen-Amor, I. et al., 2021 [87]*Eugenia uniflora*Gallocatechin, oenothein B, eugeniflorin D1, eugeniflorin D2Flavonoid, Polyphenol, TanninsEBV DNA polymeraseLee, M. et al., 2000 [92]-Gallic acid,ferrulic acidPhenolic acids,Hydroxycinnamic acidsInhibition on TPA-induced EBV activationNomura, E. et al., 2002 [93]*Ficus hispida* L.f.Chlorogenic acid, protocatechuic acid, gallic acidPhenolic acidsHydroxybenzoic acidTrihydroxybenzoic acidEBV early antigenZhang, J. et al., 2018 [94]*Lindernia crustacea* (L.) F.Muell.Luteolin-7-O-β-D--glucopyranoside, apigenin-7-O-[β-D--apiofuranosyl(1→6)-β-D--glucopyranoside]FlavonoidFlavonoidRta and ZtaWu, C.-C. et al., 2015 Wu, C.-C. et al., 2017Tsai, Y.-C. et al., 2020 [96,97,103]-Epigallocatechin-3-gallateFlavonoidLMP1, MAPKs/wt-p53 and JNKs/c-Jun pathwaysLi, H. et al., 2021 [100]-Protoapigenone, protoapigenone-1′-O-isopropyl etherFlavonoid,FlavonoidRtaTung, C.-P. et al., 2011Vágvölgyi, M. et al., 2019 [101,102]*Scutellaria barbata**Neo*-clerodane diterpenoidsDiterpenesInhibition EBV lytic replicationWu, T. et al., 2015 [103]*Euphorbia milii*Entrosane-type diterpenoidsDiterpenesInhibition EBV lytic replicationKemboi, D. et al., 2020 [104]*Litsea verticillate*LignansDiphenolic compoundsInhibition EBV lytic replicationWang, D. et al., 2016 [106]*Saururus chinensis*Manassantin B lignan*Benzodioxoles*BZLF1mTORC2Cui, H. et al., 2014, Wang, Q. et al., 2020 [105,107]*Synechococcus elongatus*Sulfated polysaccharidesGlycansReducing cell-free EBV DNAKok, Y.-Y. et al., 2011 [108]*Ankistrodesmus convolutus*Sulfated polysaccharidesGlycansReducing cell-free EBV DNAKok, Y.-Y. et al., 2011 [108]*Rhus chinensis*Moronic acidPentacyclic triterpenoidRta and ZtaEBV early antigenChang, F.-R. et al., 2010 [109]*Brazilian propolis*Moronic acidPentacyclic triterpenoidRta and ZtaEBV early antigenChang, F.-R. et al., 2010 [109]*Astragalus membranaceus*Polysaccharides
Rta and ZtaEBV early antigenGuo, Q. et al., 2014 [110]*Lawsonia inermis* L.2-hydroxy-1,4--naphthoquinoneQuinonesEBV early antigenKapadia, G.J. et al., 2013 [111]-ResveratrolNonflavonoid polyphenolRta and Zta EBV early antigenYiu, C.-Y. et al., 2010, De Leo, A. 2012 [112,113]-QuercetinFlavonoidBZLF1, BRLF1, BLLF1 and F promoterLee, M. et al., 2015Lee, M. et al., 2016 [75,114]-IsoliquiritigeninFlavonoidLMP1, LMP2, EBNA3A, EBNA1Lee, M. et al., 2015Lee, M. et al., 2016 [75,114]*Andrographis paniculata*AndrographolideTerpenoidRta and Zta EBV early antigenLin, T.-P. et al., 2008 [115]*Polygonum cuspidatum*Resveratrol, EmodinNonflavonoid polyphenol,Rta and Zta EBV early antigenYiu, C.-Y. et al., 2011,Yiu, C.-Y. et al., 2013 Yiu, C.-Y. et al., 2014 [116,117,118]CordycepsCordycepinTrihydroxyanthraquinoneLMP2A, EBNA1Ryu, E. et al., 2014 [119]


### 3.3. Natural Extracts Targeting EBV Latent Proteins and Intracellular Pathways

Research for natural extracts and compounds targeting EBV latent proteins and intracellular pathways is an active area of investigation. While there is no definitive cure or specific treatment for EBV, some natural extracts have shown promise in laboratory studies for their potential antiviral properties. Some natural extracts, studied for their capability to target EBV latent proteins and interfere with intracellular pathways, are described in Table 3. The findings reported by Ramayanti et al. provide valuable insights into the potential effects of curcumin and its analogs on EBV and associated cancers. Indeed, reduced viability of EBV-positive nasopharyngeal carcinoma cells was reported following treatment, highlighting the potential cytotoxic effect on these cancer cells. The authors also reported that curcumin and its analogs might promote apoptosis, specifically in EBV-positive cells, which could contribute to limiting the growth of cancer cells [99,120]. Intriguingly, curcumin-induced EBNA1 degradation via the proteasome-ubiquitin pathway decreased the expression of EBNA1 in HONE1 and HK1-EBV cells and inhibited the transcriptional level of EBNA1 in the HeLa cells [121]. Genipin, a natural compound extracted by *Gardenia jasminoides*, suppresses EBV infection [78] by promoting the viral lytic replication cycle in a dose-dependent manner. At 100 μM, it induces the upregulation of EBV lytic genes BNRF1, BCRF1, BLLF1, BZLF1, and EBV latent genes LMP1, LMP2A, EBNA2 in SNU719 cells and the downregulation of EBER1, EBNA3A, EBNA3C, EBNA1, and EBV lytic gene BRLF1. Unlike, the treatment of SNU719 cells with genipin at 50 μM resulted in an upregulation of EBV lytic gene BZLF1 only, while EBV latent genes LMP1, LMP2A, EBER1, EBNA2, EBNA3A, EBNA3C, EBNA1, and EBV lytic genes BNRF1, BCRF1, BLLF1, BRLF1 were downregulated [78]. These findings suggest that genipin has differential effects on EBV gene expression depending on its concentration. Briefly, at 100 μM, a broader set of genes is affected, both lytic and latent, while at 50 μM, the impact is more selective. The modulation of gene expression, especially the upregulation of lytic genes, could contribute to suppressing EBV infection. The ethanolic extract of *Polygonum cuspidatum* also inhibited the expression of LMP1, triggering the EBV-positive cells to enter apoptosis [118]. LMP1 is known for its role in promoting cell survival and preventing apoptosis, so inhibiting its expression represents a potential mechanism for triggering apoptosis in EBV-positive cells. Berberine, found in plants such as barberry (*Berberis vulgaris*) and huanglian (Coptidis rhizome), decreases the expression of EBNA1 at both mRNA and protein levels by inhibiting p-STAT3 and decreasing EBV virion production. The in vivo results of a non-toxic dose of berberine showed a decrease in tumor growth of EBV-associated NPC [99,122,123,124,125]. Triptolide produced by the thundergod vine (*Tripterygium wilfordii*) inhibits cell proliferation of EBV-positive B lymphoma cells via the downregulation of LMP1. In addition, triptolide inhibits EBNA1 expression by increasing the sensitivity for mitochondrial apoptosis in NPC [99,126,127]. A reinforcement of quercetin-mediated cytotoxicity and an enhancement of quercetin-mediated apoptosis in SNU719 cells were reported, as well as the activation of the EBV lytic gene promoter. These findings suggest that the combination of *Ganoderma lucidum* extracts and quercetin results in synergistic effects, both in terms of antitumor activity against EBVaGC cells and activation of the EBV lytic cycle [128]. Baicalein, a bioactive flavonoid compound purified from the root of *Scutellariae baicaleinsis*, inhibits the growth of Epstein-Barr virus-positive nasopharyngeal carcinoma by repressing the activity of EBNA1 [129].

It has also been illustrated that berberine-induced apoptosis by activating XAF1 and GADD45a. Berberine increases the levels of cellular reactive oxygen species and upregulates p53 by activating JNK and p38-MAPK pathways [123]. Then, p53 translocates the GADD45α (growth arrest and DNA damage-inducible alpha) protein into the nucleus and the XAF1 (X-linked inhibitor of apoptosis 1) protein into the cytosol. Furthermore, p53 upregulates PUMA, a pro-apoptotic protein that rapidly induces apoptosis via a bax- and mitochondrial-dependent pathway [99,122,123,124]. These findings suggest a complex and interconnected network of molecular events through which berberine exerts its apoptotic effect. The activation of XAF1 and GADD45a, the increase in cellular ROS levels, the upregulation of p53, and the subsequent translocation of specific proteins together contribute to the induction of apoptosis. The involvement of JNK and p38-MAPK pathways adds another layer of complexity to the signaling cascade. The information provided from the study by Hwan Hee Lee et al. indicates that quercetin and isoliquiritigenin, two flavonoids found in certain plant-based foods, have differential effects on the expression of proteins associated with apoptosis in EBV (+) human gastric carcinoma-bearing animals. The treatment with quercetin and isoliquiritigenin upregulated the expression of p53, p21, Bax, and PUMA proteins. Quercetin markedly increased the expressions of the cleaved forms of caspase 3, -9, and PARP in EBV (+) human gastric carcinoma. Quercetin’s effects were significantly more pronounced than isoliquiritigenin [114]. In Lee, M. et al., 2015, quercetin (CD_50_ 62 μM) strongly induced early apoptosis and necrosis/late apoptosis in SNU719 cells and significantly arrested the S/G2 transition of SNU719 cells, whereas isoliquiritigenin (CD_50_ 45 μM) did not impact on the cell cycle progress. Quercetin showed demethylation in cellular and viral genomes. Quercetin and isoliquiritigenin appeared to induce signal transductions linked to apoptosis, such as the MAPK/JNK pathway and MAPK/ERK pathway [75].

## 4. Discussion

EBV is a member of the herpesvirus family, capable of establishing lifelong infections in humans via both lytic and latent phases. During the latent phase, the virus resides in B lymphocytes, while the lytic phase involves productive infection in the oral mucosal epithelium. EBV encodes proteins that manipulate the host immune response, such as EBNA-1, which helps the virus to evade detection by cytotoxic T lymphocytes [130] and microRNAs that regulate gene expression both in the virus and the host [131]. EBV interferes with the host’s antigen presentation, inhibits apoptosis in infected cells, and can lead to various diseases, including cancer [132,133]. Despite causing infectious mononucleosis, EBV is associated with around 200,000 malignancies worldwide annually. Understanding these mechanisms is crucial for developing targeted treatments for EBV-related diseases. Natural antiviral compounds are substances derived from plants, fungi, or other natural sources that have demonstrated properties in preventing or treating viral infections [5]. Within this review, we provided an overview of mechanisms for natural products that have demonstrated antiviral effects towards EBV infection. 

Compounds like licorice, genipin, and EGCG prevent attachment/entry to host cells by using different mechanisms. Several compounds have demonstrated the ability to inhibit EBNA1, including curcuminoids, genipin, and baicalein, while a subset of agents functions as inhibitors of LMP1 [75,78,100,114,116,117,118,120,126,127,129]. Additionally, certain compounds not directly targeting the virus can mitigate the effects of viral infection. Compounds with robust antioxidant properties protect the cells from oxidative stress triggered by EBV replication [87,88,89,90,91,92,93,94,95]. Others exhibit antiviral effects by blocking the transcription of immediate–early genes that encode lytic proteins Rta and Zta, such as Luteolin-7-O-β-D-glucopyranoside, apigenin-7-O-[β-D-apiofuranosyl(1→6)-β-D-glucopyranoside], Protoapigenone, protoapigenone-1′-O isopropyl ether, Moronic acid, Resveratrol [96,97,98,99,100,101,102,109,112,113]. Some of them induce apoptosis, activate p53, and block cell cycle progression, limiting the growth of EBV-positive nasopharyngeal carcinoma cells [78,114,116,117,118,120,122,123,124,125,126,127,128,129].

The diversity of sources, including *Scutellaria barbata*, *Euphorbia milii*, *Saururus chinensis*, and *Litsea verticillate*, underscores the rich pool of bioactive compounds present in nature that could be further investigated for their therapeutic potential against EBV infections. Further research and clinical studies are essential to validate these findings and understand the mechanisms of action underlying the antiviral effects.

## 5. Conclusions 

This review summarizes the current information about natural compounds that inhibit or interfere with EBV infection. The current range of etiotropic drugs for EBV is limited, and some are repurposed from treatments for other viral diseases. In addition to the ongoing search for etiotropic therapy, it is crucial to discover new, potent, and safe antiviral agents that target both the virus and the infected cells. The therapeutic application of natural compounds in treating different diseases has ancient roots, and many formulations continue to be used as supportive medicines. This study sheds light on the most active molecules, discussing their efficacy in inhibiting EBV by influencing various molecular aspects of viral replication. Nevertheless, considering the extensive history of many of the mentioned compounds used as dietary supplements and their low likelihood of causing side effects, some of these phytoconstituents could serve as effective supplements to standard chemotherapy.

## Figures and Tables

**Figure 1 viruses-16-00124-f001:**
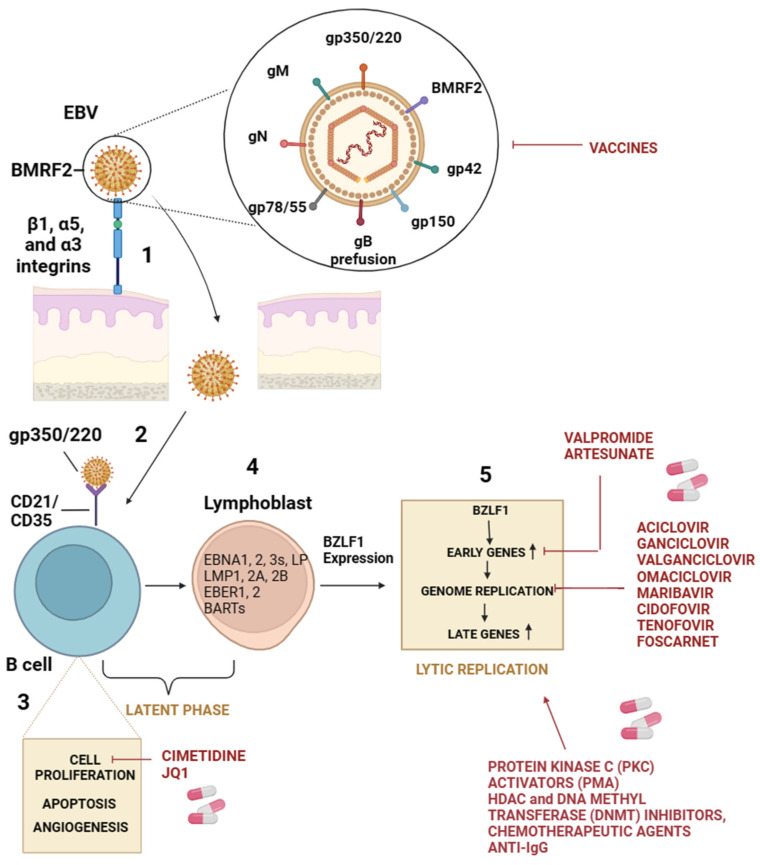
EBV infection and antiviral treatment. (1) Entry of EBV into the oropharyngeal epithelium mediated by the interaction between integrins on cell membranes and BMRF2 viral glycoprotein. (2) The infection of naïve B cells by the interaction of specific receptors on the B cell surface (CD21/CD35) with glycoproteins (gp350/220) on the viral surface. This interaction is a key step in the viral entry process and sets the stage for subsequent events in the viral life cycle within the host cell. (3) Activation of intracellular signals such as apoptosis, cell proliferation, and angiogenesis following naïve B-cell infection. (4) LCLs expressing nuclear proteins (EBNA-1, EBNA-2, EBNA-3A, EBNA-3B, EBNA-3C and EBNA-LP), membrane proteins (LMP-1, LMP-2A and LMP-2B), small RNAs (EBER1 and EBER2) and transcripts (BARTs). (5) The expression of BZLF1 is crucial to switch from latent to lytic infection. In red, antiviral drugs currently in use against EBV target several mechanisms.

**Figure 2 viruses-16-00124-f002:**
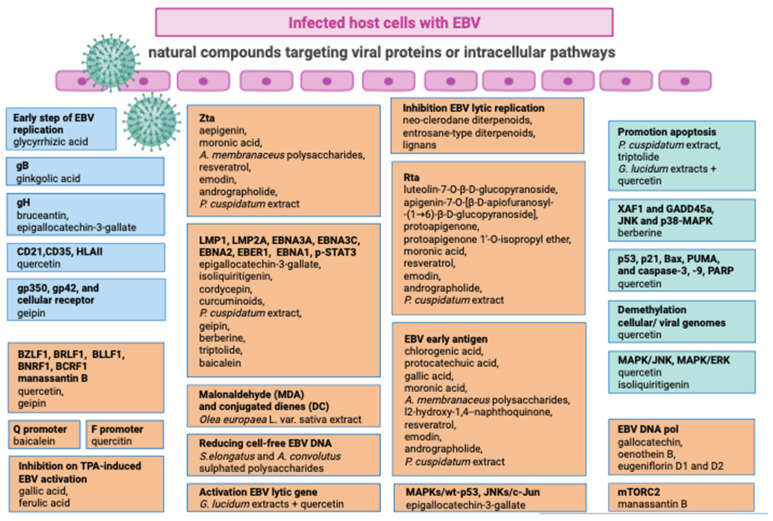
Natural compounds targeting viral proteins or intracellular pathways triggered by EBV.

**Table 1 viruses-16-00124-t001:** Natural products targeting EBV binding.

Plant	Substance	Class	Target	References
Licorice	Quercetin	Flavonoids	CD21, CD35 and HLAII EBV receptor	Lee, M. et al., 2015 [75]
*Glycyrrhizae radix*	Glycyrrhizic acid	Pentacyclic triterpenoid	Interference with early step of EBV replication	Lin, J.-C. 2003 [76]
*Gardenia jasminoides*	Genipin	Monoterpenoids	gp350 and gp42 attachment proteins, and cellular receptors	Liu, H. et al., 2013 Son, M. et al., 2015 [77,78]
-	Bruceantin	Secotriterpenoid	gH protein	Jakhmola, S. et al., 2022 [79]
Green tea	Epigallocatechin-3-gallate	Flavonoid	gH protein	Jakhmola, S. et al., 2022 [79]
*Ginkgo biloba*	Ginkgolic acid	Alkylphenol	gB protein	Borenstein, et al., 2020 [83]

**Table 3 viruses-16-00124-t003:** Natural compounds targeting EBV latent proteins and intracellular pathways.

Plant	Substance	Class	EBV Target	Host Target	References
-	Curcumin and curcuminoids	Curcuminoids/Polyphenols	EBNA1	Promotion apoptosis	Ramayanti, O. et al., 2018 [120]
*Gardenia jasminoides*	100 μM genipin	Monoterpenoids	BNRF1, BCRF1, BLLF1, BZLF1, LMP1, LMP2A, EBNA2, EBER1, EBNA3A, EBNA3C, EBNA1, BRLF1	-	Son, M. et al., 2015 [78]
*Gardenia jasminoides*	50 μM genipin	Monoterpenoids	BZLF1, LMP1, LMP2A, EBER1, EBNA2, EBNA3A, EBNA3C, EBNA1, BNRF1, BCRF1, BLLF1, BRLF1	-	Son, M. et al., 2015 [78]
*Polygonum cuspidatum*	Ethanolic extract	Alcohol	LMP1	Promotion apoptosis	Yiu, C.-Y. et al., 2013, Yiu, C.-Y. et al., 2011, Yiu, C.-Y. et al., 2014 [116,117,118]
*Berberis vulgaris*	Berberine	Protoberberine alkaloids	EBNA1, p-STAT3	XAF1 and GADD45a, JNK and p38-MAPK	Wang, C. et al., 2017, Park, G.B. et al., 2016, Tsang, C.M. et al., 2013, Zhou, F. et al., 2020 [122,123,124,125]
Coptidis rhizome	Berberine	Protoberberine alkaloids	EBNA1, p-STAT3	XAF1 and GADD45a, JNK and p38-MAPK	Wang, C. et al., 2017, Park, G.B. et al., 2016, Tsang, C.M. et al., 2013, Zhou, F. et al., 2020 [122,123,124,125]
*Tripterygium wilfordii*	Triptolide	Epoxide	LMP1, EBNA1	Promotion apoptosis	Zhou, H. et al., 2015, Zhou, H. et al., 2018 [126,127]
*Ganoderma lucidum*	*Ganoderma lucidum* extracts + quercetin	Agaricomycetes	Activation of EBV lytic gene	Promotion apoptosis	Sora, H. et al., 2019 [128]
*Scutellariae baicaleinsis*	Baicalein	Flavonoid	EBNA1 Q-promoter	-	Zhang, Y. et al., 2018 [129]
-	Quercetin	Flavonoid	-	p53, p21, Bax, PUMA, and caspase 3, -9, PARP, demethylation cellular/ viral genomes, MAPK/JNK MAPK/ERK	Hwan, H.L. et al., 2016 [114]
-	Isoliquiritigenin	Flavonoid	-	MAPK/JNK MAPK/ERK	Lee, M., et al., 2015 [75]

## Data Availability

Not applicable.

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
