# Peer review of "Update of Natural Products and Their Derivatives Targeting Epstein–Barr Infection"

_viruses, 2024, doi:10.3390/v16010124_

Round 1

Reviewer 1 Report

Comments and Suggestions for Authors

A very comprehensive review, written and presented to a high standard.

Introduction - lines 29-30. The current statement incorrect. Suggest

"EBV commonly infects people in developed and developing countries. Most cases are asymptomatic although infectious mononucleosis can manifest in individuals, particularly as the age of infection increases."

Lines 214-215. "Sporadic BLs are usually EBV-negative."  Please add reference.

Lines 254-255. There seems to be some confusion here. Do the authors mean relapsing/reactivating or relapsing/remitting? In either case, I am not aware of a definitive link between EBV lytic activity and the relapsing state. The authors should carefully recheck their statement.

Lines 261-265. I do not agree with this statement. The processes described may contribute to MS as environmental factors but they do not account for the genetic linkages associated with MS.

Tables - Need to cite the reference numbers (so they can be located in the references section) as well as the authors and date

Author Response

We sincerely thank the reviewer for the comments, which helped us in improving the revised version of the manuscript. Please find below a point-by-point description that includes the original reviewer's bold comments and the responses in regular typeface. A revised version of the manuscript has been resubmitted.

Introduction - lines 29-30. The current statement incorrect. Suggest "EBV commonly infects people in developed and developing countries. Most cases are asymptomatic although infectious mononucleosis can manifest in individuals, particularly as the age of infection increases." We thank the reviewer for the suggestion. We edited the manuscript accordingly as reported in lines 30-32. 

  1. Lines 214-215. "Sporadic BLs are usually EBV-negative."  Please add reference. We edited the manuscript and removed the section.
  2. Lines 254-255. There seems to be some confusion here. Do the authors mean relapsing/reactivating or relapsing/remitting? In either case, I am not aware of a definitive link between EBV lytic activity and the relapsing state. The authors should carefully recheck their statement. We edited the manuscript and removed the section.

  1. Lines 261-265. I do not agree with this statement. The processes described may contribute to MS as environmental factors but they do not account for the genetic linkages associated with MS. We appreciate the reviewer's feedback and we have revised the introduction as outlined in the manuscript. We acknowledge the concerns raised by the referee, recognizing that multiple factors contribute to the complex and multifactorial aetiology of multiple sclerosis. These factors encompass known genetic susceptibility elements, primarily associated with immune system regulation, environmental influences such as infectious agents, insufficient sun exposure and vitamin D, smoking and obesity. The suspicion of infectious agents in MS etiology dates to its classification in the late 1800s. The heterogeneity and dynamic evolution of the disease over a patient's lifetime, as well as within MS lesions, have complicated the efforts to identify a singular infectious agent as a consistent trigger for the disease. Nevertheless, a growing body of epidemiological, serological and virological evidences supports the role of the Epstein-Barr virus in the etiology of MS. Recent extensive population-based studies have strengthened the association, indicating that EBV infection likely constitutes a prerequisite for the development of the disease (Soldan, S.S.; Lieberman, P.M. Epstein–Barr Virus and Multiple Sclerosis. Nat Rev Microbiol 2023, 21, 51–64, doi:10.1038/s41579-022-00770-5.) . Therefore, we have modified the sentence to maintain the idea of the crucial role of the Epstein-Barr virus in the etiology of multiple sclerosis and added the reference 3 to reinforce this perspective.
  2. Tables - Need to cite the reference numbers (so they can be located in the references section) as well as the authors and date. We thank the reviewer for the suggestion. We edited the manuscript accordingly.

Reviewer 2 Report

Comments and Suggestions for Authors

I have reviewed the paper by Pennisi et al.

The article is extensive. In fact, I would say it its extremely long. I suggest author keep a more focused approach with a shorter Introduction and Section 2, and then deal with the actual topic which starts on section 3. Authors need to reduce the content by at least 75%, as they deal with entries that are outside the main theme, and will be unrelated to what the action of the therapeutic compounds.

Although the English is fine, they need to keep length of paragraphs shorter. Rather than enumerating and describing methods of each of the 220 studies in the reference list, they need to write a short sentence with the strong message of each study. That would make this Review readable, lighter and more digestible.

Author Response

We sincerely thank the reviewer for the comments, which helped us in improving the revised version of the manuscript. Please find below a point-by-point description that includes the original reviewer's bold comments and the responses in regular typeface. A revised version of the manuscript has been resubmitted.

  1. I have reviewed the paper by Pennisi et al. The article is extensive. In fact, I would say it its extremely long. I suggest author keep a more focused approach with a shorter Introduction and Section 2, and then deal with the actual topic which starts on section 3. Authors need to reduce the content by at least 75%, as they deal with entries that are outside the main theme, and will be unrelated to what the action of the therapeutic compounds. Although the English is fine, they need to keep length of paragraphs shorter. Rather than enumerating and describing methods of each of the 220 studies in the reference list, they need to write a short sentence with the strong message of each study. That would make this Review readable, lighter and more digestible.

We are grateful to the reviewer for the positive comments. We revised the manuscript, by reducing the content and eliminating some paragraphs. The extensive editing has enhanced the paper's readability.

Reviewer 3 Report

Comments and Suggestions for Authors

The review appeared to be even more comprehensive than one could expect on the base of its title and Abstract. The manuscript describes in details EBV potential molecular targets for antivirals, cellular tropism, as well as available synthetic drugs and natural products. Really, it's not "update" since aciclovir had been known from 1970s (more than 50 years) and green tea extract even more.

Short Abstract is full of contradictions. Thus, the statement that EBV "infects epithelial cells and establishes latent infection" from the first sentence may cause misunderstanding since "acute infection" and B cells are mentioned later. Moreover, multiple uncommon abbreviations (such as BZLF, BRLF in the Abstract and throughout the text are not explained and Abbreviation list is missing. Some of the abbreviations are highlighted in yellow in the attached file.

Unfortunately, some scientific terms are not always used accordingly.

Line 33. "copy number of it genome".

Perhaps, concentration? Genome-equivalents per cell? Or in unit of volume? How viral proteins can calculate or estimate "copy number of it genome"? Recognition and binding with virions?

Line 43.

Gene expression stands for transcription with subsequent translation. But what means protein expression?

Line 48. Corticosteroids are known to be a class of steroid hormones with anti-inflammatory properties but not only "immunomodulators". Their mechanisms of action are more complex. 

Lines 51-53. "Indeed, the persistence of the viral genome, as well as the uncontrolled immune stimulation, produces oxidative stress and results in various neuropathies, including multiple sclerosis"

EBV naked DNA cannot persist. Usually virus persistence inside infected cells does not result in strong immune response, especially "uncontrolled" cytokine storm with subsequent adaptive immunity. Besides that, multiple sclerosis is autoimmune disease.

Line 61. "various steps of viral infection, such as infection..."

Please, rephrase to make it clear.

Natural compounds may also be highly toxic. For example, snake venoms, bee venom (peptide melittin) etc.

Line 65. Differences of toxicity in vitro and in vivo should be explained at levels of proteins, specialized cells, organs and whole organisms with excretion systems.

Line 79. Abbreviations are not explained.

Line 88. What mean "shell particles"? Their chemical composition and physical properties.

Line 89. "Entry receptors" - please, explain.

Line 91. EBV "accept glycoproteins". What viral structural surface proteins or glycoproteins can recognize and specifically bind with cellular glycoproteins?

Line 103. "Encapsidation" of virions? Additional capsid along with nucleocapsid?

Section 2.4. Antiviral therapy.

Line 274. Vaccines are not antiviral therapeutic agents. They are supposed to prevent the infections.

Lines 284-288.

Differences between two first groups of nucleoside and nucleotide analogs (i and ii) remain unclear. Based on their definition they are chemical analogs of nucleosides and nucleotides and, consequently, can interfere with natural nucleosides and nucleotides necessary for both viral and cellular DNA replication. Therefore, drugs from  both groups should be highly toxic for dividing cells. According to suggested classification they belong to different groups. Why?

Line 323. Are Fe ions cofactors of EBV-specific enzymes? Why "iron chelators" are suggested for treatment despite their evident side effects especially in vivo?

Lines 356-369. Description of vaccine candidates seems to be non-relevant to the manuscript. They are neither "natural" nor antivirals.

Line 407. "antiviral chemotherapeutic agents"

Commonly, chemotherapy of cancer but not viral infections.

I really doubt about specific targeting of natural products towards EBV proteins (sections 3.1 and 3.2). Additional column in tables 1-3 with classes of natural products would be helpful. As one can see most of them are polyfunctional plant polyphenols that non-specifically protect plants from infections and are anti-oxidants.

Unfortunately, discussion does not fit for Abstract and main text, too.

Line 678  EBV "primarily infects B lymphocytes" - line 12 "infects epithelial cells".

Line 706 "Strikingly, there are currently no vaccines or antiviral drugs 706 available to prevent or treat EBV infection".

Epidemiology of EBV in the world in dynamics is currently absent in the manuscript but may permit to compare EBV infection risks and the need for vaccines (including both development, production and medical requirements) thus to find the answer.

Conclusion is highly desirable. WHO strongly recommend etiotropic therapy. Natural products are polyfunctional and unspecific with possible allergic complications.

Comments on the Quality of English Language

Abbreviation list should be included in the manuscript.

Scientific terms must be used properly.

Moderate editing of English language is recommended.

Author Response

We sincerely thank the reviewer for the comments, which helped us in improving the revised version of the manuscript. Please find below a point-by-point description that includes the original reviewer's bold comments and the responses in regular typeface. A revised version of the manuscript has been resubmitted.

.

  1. The review appeared to be even more comprehensive than one could expect on the base of its title and Abstract. The manuscript describes in details EBV potential molecular targets for antivirals, cellular tropism, as well as available synthetic drugs and natural products. Really, it's not "update" since aciclovir had been known from 1970s (more than 50 years) and green tea extract even more. We agree with the reviewer. We revised the manuscript, by reducing the content and eliminating some paragraphs. The extensive editing has enhanced the paper's readability.
  2. Short Abstract is full of contradictions. Thus, the statement that EBV "infects epithelial cells and establishes latent infection" from the first sentence may cause misunderstanding since "acute infection" and B cells are mentioned later. Moreover, multiple uncommon abbreviations (such as BZLF, BRLF in the Abstract and throughout the text are not explained and Abbreviation list is missing. Some of the abbreviations are highlighted in yellow in the attached file. We thank the reviewer for the observation. We re-edited the sentence as follows: “Epstein Barr (EBV) is a human γ-herpesvirus that undergoes both a productive (lytic) cycle and a non-productive (latent) phase. The virus establishes enduring latent infection in B lymphocytes and productive infection in the oral mucosal epithelium.” Besides, we added Appendix A with a list of abbreviations and acronyms used in the paper.

  1. Unfortunately, some scientific terms are not always used accordingly. Line 33. "copy number of it genome". Perhaps, concentration? Genome-equivalents per cell? Or in unit of volume? How viral proteins can calculate or estimate "copy number of it genome"? Recognition and binding with virions? We thank the reviewer for the observation. We re-edited the introduction as reported in the manuscript.
  2. Line 43. Gene expression stands for transcription with subsequent translation. But what means protein expression? We thank the reviewer for this comment. We re-edited the introduction as reported in the manuscript.
  3. Line 48. Corticosteroids are known to be a class of steroid hormones with anti-inflammatory properties but not only "immunomodulators". Their mechanisms of action are more complex. We thank the reviewer for the observation. We re-edited the introduction as reported in the manuscript.
  4. Lines 51-53. "Indeed, the persistence of the viral genome, as well as the uncontrolled immune stimulation, produces oxidative stress and results in various neuropathies, including multiple sclerosis" . EBV naked DNA cannot persist. Usually virus persistence inside infected cells does not result in strong immune response, especially "uncontrolled" cytokine storm with subsequent adaptive immunity. Besides that, multiple sclerosis is autoimmune disease. We appreciate the reviewer's feedback and have revised the introduction as outlined in the manuscript. We acknowledge the concerns raised by the reviewer, recognizing that multiple factors contribute to the complex and multifactorial etiology of multiple sclerosis. These factors encompass known genetic susceptibility elements, primarily associated with immune system regulation, as well as environmental influences such as infectious agents, insufficient sun exposure and vitamin D, smoking, and obesity. The suspicion of infectious agents in MS etiology dates to its classification in the late 1800s. The heterogeneity and dynamic evolution of the disease over a patient's lifetime, as well as within MS lesions, have complicated the efforts to identify a singular infectious agent as a consistent trigger for the disease. Nevertheless, a growing body of epidemiological, serological, and virological evidence supports the role of Epstein-Barr virus in the etiology of MS. Recent large population-based studies have strengthened the association, indicating that EBV infection likely constitutes a prerequisite for the development of the disease. Therefore, we have modified the sentence to maintain the idea of the crucial role of the Epstein-Barr virus in the etiology of multiple sclerosis and added the reference 3 to reinforce this perspective.
  5. Line 61. "various steps of viral infection, such as infection..." Apologies for this error that has been corrected as follows: “Numerous natural compounds have been screened and identified as viral inhibitors targeting various steps of viral replication, such as the entry, uncoating, genome late gene expression, assembly, secretion, and cellular processes required for virion production”.
  6. Natural compounds may also be highly toxic. For example, snake venoms, bee venom (peptide melittin) etc. Line 65. Differences of toxicity in vitro and in vivo should be explained at levels of proteins, specialized cells, organs and whole organisms with excretion systems. We understand the reviewer's concerns. However, in this statement, we highlight that while utilizing natural products as medicines offers certain advantages, there are also various drawbacks. One notable disadvantage is the substantial disparity between the in vitro cellular approach to toxicity testing and the clinical approach, potentially contributing to the failure of drug approval.

  1. Line 79. Abbreviations are not explained. We thank the reviewer for the observation. We added Appendix A with a list of abbreviations and acronyms used in the paper.

  1. Line 88. What mean "shell particles"? Their chemical composition and physical properties. We re-edited the sentence: “The EBV virion, with a diameter of about 150–170 nm, is composed of an icosahedral nucleocapsid with 162 capsomers surrounded by an envelope.”
  2. Line 89. "Entry receptors" - please, explain. Apologies for this error. . We re-edited the sentence: “Viral surface glycoproteins are responsible for recognition and binding to cellular receptors and for consequent membrane fusion.”
  3. Line 91. EBV "accept glycoproteins". What viral structural surface proteins or glycoproteins can recognize and specifically bind with cellular glycoproteins? We re-edited the sentence: “EBV possesses a wide range of glycoproteins, such as gp350, gp42, gH, gL, and gB”
  4. Line 103. "Encapsidation" of virions? Additional capsid along with nucleocapsid? We re-edited the sentence:“The interaction between the EBV envelope glycoprotein, gp350/220, and CD21/CD35 is responsible for the binding, and triggers a signaling cascade allowing the penetration of the virion in B cells”.
  5. Section 2.4. Antiviral therapy. Line 274. Vaccines are not antiviral therapeutic agents. They are supposed to prevent the infections. Lines 284-288. Differences between two first groups of nucleoside and nucleotide analogs (i and ii) remain unclear. Based on their definition they are chemical analogs of nucleosides and nucleotides and, consequently, can interfere with natural nucleosides and nucleotides necessary for both viral and cellular DNA replication. Therefore, drugs from  both groups should be highly toxic for dividing cells. According to suggested classification they belong to different groups. Why? Line 323. Are Fe ions cofactors of EBV-specific enzymes? Why "iron chelators" are suggested for treatment despite their evident side effects especially in vivo? Lines 356-369. Description of vaccine candidates seems to be non-relevant to the manuscript. They are neither "natural" nor antivirals. Line 407. "antiviral chemotherapeutic agents". Commonly, chemotherapy of cancer but not viral infections. We appreciate the reviewer's feedback. Section 2.4 has been removed as it was non-relevant for the manuscript's topic.
  6. I really doubt about specific targeting of natural products towards EBV proteins (sections 3.1 and 3.2). Additional column in tables 1-3 with classes of natural products would be helpful. As one can see most of them are polyfunctional plant polyphenols that non-specifically protect plants from infections and are anti-oxidants.
  7. We understand the reviewer's concerns. We have modified the tables by incorporating an additional column indicating the class of natural compounds. This modification contributes to improved readability and positively influences the overall quality of the manuscript.
  8. Unfortunately, discussion does not fit for Abstract and main text, too. Line 678  EBV "primarily infects B lymphocytes" - line 12 "infects epithelial cells". Line 706 "Strikingly, there are currently no vaccines or antiviral drugs 706 available to prevent or treat EBV infection". We appreciate the reviewer's feedback. We re-edited the Discussion.
  9. Epidemiology of EBV in the world in dynamics is currently absent in the manuscript but may permit to compare EBV infection risks and the need for vaccines (including both development, production and medical requirements) thus to find the answer. We appreciate the referee's concerns and acknowledge the significance of the epidemiological aspect of the infection. However, it is essential to note that the manuscript's focus does not encompass this topic. As evident from the extensive revisions made, we have eliminated elements deemed superfluous in accordance with the feedback from various reviewers. Therefore, the discussion of the epidemiological aspect is currently considered non-essential.
  10. Conclusion is highly desirable. WHO strongly recommend etiotropic therapy. Natural products are polyfunctional and unspecific with possible allergic complications.

We understand the reviewer concerns. We recognize the importance of antiviral drugs and monoclonal antibodies targeting EBV directly. However, as we wrote in the “conclusion section”, the current range of etiotropic drugs for EBV is limited, and some are repurposed from treatments for other viral diseases. In addition to the ongoing search for etiotropic therapy, there is a crucial need to discover new, potent, and safe antiviral agents that target both viral particles and the cellular response to viral infection. The therapeutic application of natural compounds in treating various diseases has ancient roots and many formulations continue to be used as supportive medicines. This study sheds light on the most active molecules, discussing their efficacy in inhibiting EBV by influencing various molecular aspects of viral infection. Nevertheless, considering the extensive history of many of the mentioned natural compounds used as dietary supplements and their low likelihood of causing side effects, some of these phytoconstituents could serve as effective supplements to standard chemotherapy.

  1. Abbreviation list should be included in the manuscript. We thank the reviewer for the suggestion. We added a list of abbreviations as reported in the manuscript.

  1. Scientific terms must be used properly. We thank the reviewer for the observation and we trust that the extensive editing has enhanced the paper's readability.

  1. Moderate editing of English language is recommended. We express our appreciation to the reviewer for his valuable suggestion. The manuscript underwent meticulous analysis by a skilled linguist, whom we included as an author in the manuscript, and we trust that the extensive editing has enhanced the paper's readability.

Round 2

Reviewer 2 Report

Comments and Suggestions for Authors

A great improvement from the previous version.

Articles is acceptable for publication.